# Ectopic Expression of *AeNAC83*, a NAC Transcription Factor from *Abelmoschus esculentus*, Inhibits Growth and Confers Tolerance to Salt Stress in *Arabidopsis*

**DOI:** 10.3390/ijms231710182

**Published:** 2022-09-05

**Authors:** Xuan Zhao, Tingting Wu, Shixian Guo, Junling Hu, Yihua Zhan

**Affiliations:** The Key Laboratory for Quality Improvement of Agricultural Products of Zhejiang Province, College of Advanced Agricultural Sciences, Zhejiang A & F University, Hangzhou 311300, China

**Keywords:** okra, salt stress, growth, NAC transcription factor, flavonoid, photosynthesis

## Abstract

NAC transcription factors play crucial roles in plant growth, development and stress responses. Previously, we preliminarily identified that the transcription factor *AeNAC83* gene was significantly up-regulated under salt stress in okra (*Abelmoschus esculentus*). Herein, we cloned the nuclear-localized AeNAC83 from okra and identified its possible role in salt stress response and plant growth. The down-regulation of *AeNAC83* caused by virus-induced gene silencing enhanced plant sensitivity to salt stress and increased the biomass accumulation of okra seedlings. Meanwhile, *AeNAC83*-overexpression *Arabidopsis* lines improved salt tolerance and exhibited many altered phenotypes, including small rosette, short primary roots, and promoted crown roots and root hairs. RNA-seq showed numerous genes at the transcriptional level that changed significantly in the *AeNAC83*-overexpression transgenic and the wild *Arabidopsis* with or without NaCl treatment, respectively. The expression of most phenylpropanoid and flavonoid biosynthesis-related genes was largely induced by salt stress. While genes encoding key proteins involved in photosynthesis were almost declined dramatically in *AeNAC83*-overexpression transgenic plants, and NaCl treatment further resulted in the down-regulation of these genes. Furthermore, DEGs encoding various plant hormone signal pathways were also identified. These results indicate that AeNAC83 is involved in resistance to salt stress and plant growth.

## 1. Introduction

Salinity is one of the most detrimental abiotic stresses which limits plant growth and reduces biomass and grain yield severely worldwide. Salt stress impairs the productivity of plants by affecting cell growth and metabolic process, causing irreversible damage to seed germination, seedling growth and crop yield [1]. To counter the adverse effects of environmental stress, plants have evolved complex mechanisms to cope with salt stress at both physiological and biochemical levels [2]. The improvement of morphological structure and physiological metabolism level is controlled by the expression of stress-response genes, and transcription factors (TFs) play a prominent part in regulating these genes. Numerous TFs, including NAC TFs, have been identified to be involved in plant growth and salt stress response. 

The plant-specific NAC TF family is one of the largest TF families in plants, named after the initials of the *NAM* (no apical meristem) gene from *Petunia hybrida* [3], the *ATAF1/2* [4] and *CUC2* (cup-shaped cotyledon) [5] from *Arabidopsis thaliana*. In 1996, Souer et al. [3] cloned the first NAC transcription factor gene *NAM* from *Petunia*, which affects the formation and differentiation of *Petunia* apical meristem. Subsequently, NAC TFs were successively identified in *Arabidopsis* [6], rice [7], soybean [8], tomato [9] and other species. The N-terminal of all NAC TFs contains a conserved NAC domain composed of about 150 amino acid residues, and the C-terminal contains a highly variable transcriptional activation region (TAR) [6]. 

Many studies have shown that NAC TFs not only play an important role in plant growth and development, such as secondary wall formation [10,11,12], leaf senescence [13], and lateral root development [14,15], but also participate in response to abiotic stresses [16,17]. NAC TFs respond to salt stress mainly through maintaining intracellular Na^+^ and K^+^ concentrations and the relative homeostasis of the intracellular environment by activating the expression of stress response genes and protecting the stability of cell structure by enhancing the accumulation of osmoregulation substances such as soluble sugar, proline, and betaine. For example, compared with WT, transgenic rice overexpressing *ONAC022* with lower Na^+^ accumulation and transpiration rate, and increased free proline and soluble sugar content, enhanced salt tolerance significantly [18], which were also observed in transgenic rice overexpressing *ONAC009* and *ONAC058* [19,20]. Wheat TaNAC29 improved plant salt tolerance by inducing the expression of stress-related genes, enhancing the antioxidant system, and scavenging reactive oxygen species [21]. Under salt stress, *NAC13*-overexpressing transgenic poplar showed enhanced salt tolerance and the *NAC13*-suppressing plants increased sensitivity to salt stress [22]. 

Okra (*Abelmoschus esculentus* L.), a medicinal and edible plant, has attracted increasing attention worldwide [23,24]. However, no molecular characterization of any NAC family member’s impact on okra growth and salt stress response has been reported. In our previous study, an NAC TF gene *AeNAC83* from okra was up-regulated and exposed to salt stress [25]. In the present work, we cloned the coding sequence of AeNAC83 by using full-length isoforms obtained in okra and silenced the gene by the whole plant virus-induced gene silencing (VIGS). The *AeNAC83*-silenced okra seedlings exhibited a significantly higher biomass accumulation under normal conditions and were more sensitive to salt stress than that in Mock plants. Moreover, we further investigated its functions in transgenic *Arabidopsis* overexpressing *AeNAC83*. The overexpression of *AeNAC83* improved salt resistance and inhibited plant growth. According to the transcriptome analysis, the improved resistance and inhibited growth were related to the up-regulation of flavonoid biosynthesis-related genes and the down-regulation of photosynthesis-related genes in transgenic plants. The above work will provide important resources for molecular breeding of plant stress resistance.

## 2. Results

### 2.1. Isolation and Characterization of AeNAC83

We first cloned the coding sequence of AeNAC83 from okra, which encoded a protein of 255 amino acids. Then, multiple sequence alignment was performed with other NAC TFs from *Arabidopsis* and other species. AeNAC83 has a high homology with other NAC members, containing one NAC domain with five conserved regions (Figure 1A). In order to reveal the evolutionary relationship between AeNAC83 and other NAC TFs, a phylogenetic tree was constructed. All the selected NAC members were clustered into two distinct subgroups, and AeNAC83 belonged to the ATAF subgroup (Figure 1B).

To determine the subcellular location of AeNAC83, the 35Sp:: *AeNAC83*:GFP fusion protein was constructed and transiently expressed in tobacco epidermal cells. Fluorescence analysis showed that the control empty vector was expressed in both the nucleus and cytoplasm, while the protein was only localized in the nucleus (Figure 2A), indicating that AeNAC83 was localized in the nucleus.

Our previous analysis showed that salt stress induced the expression of *AeNAC83*. To further confirm whether AeNAC83 participates in the salt resistance, the two-week-old okra seedlings were irrigated with 300 mM NaCl and sampled at 1, 3, 5 and 7 days after treatment, and then the transcription level of the *AeNAC83* gene in the second true leaf of the seedlings was determined by qRT-PCR. We found that compared with the control, the expression of *AeNAC83* was up-regulated at three time points after salt treatment, and the expression was highest on the first day, then gradually decreased (Figure 2B). The result suggests that AeNAC83 may play a regulatory role in salt stress in okra.

### 2.2. Performance of AeNAC83-Silenced Plants under Salt Stress

The VIGS method was used to assess the *AeNAC83* gene function. We constructed the pTRV2-*NAC83* vector with a 336-bp fragment and then inoculated the *Agrobacterium* mixture containing the pTRV2-*NAC83* vector and pTRV1 into the cotyledons of okra. Negative control was performed by using empty vectors pTRV1 and pTRV2. The *AeNAC83* mRNA levels of newly grown young true leaves at 25-days post-inoculation were detected to assess the efficiency of VIGS. As shown in Figure 3A, the mRNA levels of AeNAC83 in silenced plants decreased dramatically compared with the control, accounting for about 60% of the control, indicating that the gene was partially silenced. 

To determine the roles of AeNAC83 in salt stress and plant growth, the *AeNAC83*-silenced okra seedlings produced by VIGS were irrigated with water or 300 mM NaCl solution for 7 days. Under normal conditions, the total leaf fresh weight of *AeNAC83*-silenced seedlings was significantly higher than Mock plants, indicating that AeNAC83 may be involved in plant growth. After the 7-day treatment, the seedling growth was inhibited obviously (Figure 3B), and leaf fresh weight of *AeNAC83*-silenced plants decreased significantly than that of Mock plant (Figure 3C). The total chlorophyll content in *AeNAC83*-silenced plants was not significantly different from that in the control before salt treatment, whereas after salt treatment the chlorophyll content increased in the Mock plants (Figure 3D). The results showed that *AeNAC83*-silenced plants were more sensitive to NaCl treatment, indicating that AeNAC83 may play a positive regulatory role in salt stress. 

### 2.3. Overexpression of AeNAC83 in Arabidopsis Inhibits Plant Growth and Improves Salt Tolerance

To further investigate the role of AeNAC83 in salt stress and growth, over-expression of *AeNAC83* gene transgenic Arabidopsis plants were produced. The lines OX3 and OX7 with high expression of *AeNAC83* were selected for further phenotypic analysis from at least 10 homozygous transgenic lines (Figure 4A).

The *Arabidopsis thaliana* ecotype Columbia (wild-type, WT) and transgenic seedlings were grown in 1/2 MS medium for 4 days and then transplanted to medium with 0, 120 or 150 mM NaCl for 10 days to observe the phenotypic difference. Under normal conditions, significant differences existed in the growth between the WT and transgenic plants. Compared with WT, *AeNAC83*-overexpression transgenic plants exhibited many altered phenotypes (Figure 4B), including small rosette, short primary roots and promoted crown roots and root hairs (Figure 4E). The fresh weight (Figure 4C) and primary root length (Figure 4D) of the transgenic plants were significantly declined than those of wild-type plants. After NaCl treatment for 10 days, better performance was observed for transgenic lines than WT. Most WT leaves showed albinistic symptoms, and the number of crown roots of transgenic lines was greater than in control (Figure 4D). The fresh weight of the transgenic lines was greater than that of WT exposed to salt stress, and there was no obvious difference in primary root length between WT and OX7 transgenic lines. Together, these results suggest that over-expression of *AeNAC83* inhibited plant growth and enhanced tolerance to salt stress. 

### 2.4. Identification and Functional Enrichment Analysis of Differential Expression Genes (DEGs) under Salt Stress between the AeNAC83-Overexpression Transgenic and the Wild Arabidopsis

To understand the molecular mechanism of AeNAC83 in plant growth and salt stress response, RNA was extracted from *AeNAC83*-overexpression transgenic (OX3) and the wild (WT) *Arabidopsis* treated with 120 mM NaCl (12 samples, three replicates for each treatment) and RNA-seq was carried out using Illumina sequencing platform. The Pearson correlation coefficient heat map showed that the samples had good repeatability (Appendix A). A total of 245.54 M reads was obtained, with a total of 73.35 Gb clean data. Clean data of each sample reached 5.74 Gb, and the percentage of Q30 was at least 93.91% (Appendix A). We assembled and quantified reads compared with HISAT2 using StringTie and performed fragments per kilobase of transcript per million fragments mapped (FPKM) conversion to analyze gene expression level. The screening of threshold for DEGs was Padj < 0.05 and |log2FoldChange| > 1. The hierarchical clustering analysis of DEGs under different experimental conditions was shown in Figure 5A. The volcanoplots represent the overall distribution of DEGs (Figure 5B). Compared with WT-CK, there were 1917 DEGs in OX3-CK, including 1259 up- and 658 down-regulated genes, and there were 4285 DEGs in WT-N including 2098 up- and 2187 down-regulated genes. Compared with OX3-CK, 1360 DEGs were found in OX3-N, including 692 up and 668 down-regulated genes. Compared with WT-N, there were 1537 DEGs in OX3-N, including 775 up- and 762 down-regulated genes (Figure 5C, Appendix A). Venn diagram showed that the number of DEGs between “WT-CK vs. OX3-CK”, “WT-CK vs. WT-N”, “OX3-CK vs. OX3-N”, and “WT-N vs. OX3-N” were 1917, 4285, 1360 and 1537, respectively. A total of 105 common DEGs were identified in the four comparative groups (Figure 5D). 

### 2.5. Identification of DEGs Involved in the Phenylpropanoid and Flavonoid Biosynthesis Pathways

KEGG pathway enrichment analysis was performed to interpret the functions of these DEGs. The top three DEGs participated in “MAPK signaling pathway-plant”, “plant-pathogen interaction” and “phenylpropanoid biosynthesis” in WT-CK vs. OX3-CK and WT-CK vs. WT-N comparison groups (Figure 6A,B). The top three DEGs participated in “phenylpropanoid biosynthesis”, “MAPK signaling pathway”, and “starch and sucrose metabolism” in OX3-CK vs. OX3-N comparison group (Figure 6C). The top three DEGs participated in “phenylpropanoid biosynthesis”, “circadian rhythm-plant”, and “starch and sucrose metabolism” in WT-N vs. OX3-N comparison group (Figure 6D). 

The phenylpropanoid biosynthesis pathway was always found to contain significant enrichment of DEGs for all comparisons (Figure 6). Through integration, 13 gene families involved in phenylpropanoid biosynthesis were identified (Figure 7A), including 129 DEGs. The POD family was the most represented (49 DEGs) among these 13 gene families, whereas the C4H, F5H, and CSE family were the least represented, with only 1 DEG (Figure 7A). The phenylpropanoid biosynthesis pathway provides the precursors for the flavonoid biosynthesis pathway [26]. Hence, we analyzed the DEGs involved in flavonoid biosynthesis and 10 gene families were identified, among which the HCT family was the most represented (7 DEGs). The expression levels of most DEGs were up-regulated by NaCl treatment, except for 2 *CHI* genes and 1 *HCT* gene (Figure 7B). In addition, under normal conditions, nearly half of the flavonoid biosynthesis-related DEGs were significantly induced by overexpression of *AeNAC83*. *CHI*, *F3H*, *DFR*, and *ANS* are crucial structural genes involved in the anthocyanin synthesis pathway. Anthocyanins, a class of flavonoids distributed ubiquitously in the plant, play important roles in the growth and development of plants and stress response. The content of anthocyanins in OX3 was significantly higher than that of WT before salt treatment. Salt stress induced the accumulation of anthocyanins in WT, while there was no significant change in anthocyanin content in OX3 (Appendix A).

### 2.6. Identification of DEGs That Participate in Photosynthesis and Hormone Signal Transduction Pathways 

Thylakoid membrane photosynthetic complexes consist of photosystem II (PSII), cytochrome b6f, photosystem I (PSI), light-harvesting antenna complexes, and ATP synthase [27]. Compared with WT-CK, genes encoding key proteins involved in these complexes were almost greatly decreased in *AeNAC83*-overexpression transgenic (OX3) *Arabidopsis*, and NaCl treatment further resulted in the down-regulation of these genes, except for 1 *PsbP* gene of PSII and 1 *gamma* gene of ATP synthase (Figure 8). Chlorophyll is one of the most important pigments in higher plants, which is an important substance for photosynthesis. The total chlorophyll content was similar between OX3 and WT before salt treatment. After salt treatment, the chlorophyll content decreased dramatically in both WT and OX3 plants (Appendix A).

In addition, the abundance of hormone-related DEGs changed significantly under salt stress. A large number of genes encoding AUX1, AUX/IAA, ARF, GH3 and SAUR for the auxin signaling pathway were identified, including 2 *AUX1* genes, 8 *AUX/IAA* genes, 1 *ARF* gene, 13 *GH3* genes and 21 *SAUR* genes; 1 *CRE1* gene, 2 *AHP* genes, 11 *B*-*ARR* genes and 2 *A*-*ARR* genes for the cytokinin signaling pathway were obtained; 3 *GID1* genes, 8 *DELLA* genes and 13 *TF* genes were identified in the GA signaling pathway; DEGs involved in ABA signaling pathway were identified, including 2 *PYR* genes, 8 *PP2C* genes, 4 *SnRK2* genes and 4 *AREB/ABF* genes; 1 *ETR*, 2 *CTR1* genes, 4 *SIMKK* genes, 1 *EBF1/2* gene and 4 *ERF1/2* genes for ET signaling pathway were analyzed; 9 *BAK1* genes, 4 *BRI1* genes, 2 *BSK* genes, and 1 *TCH4* gene related to the BR signaling pathway were identified; there were 5 *JAZ* and 9 *MYC2* DEGs in the JA signaling pathway; for the SA signaling pathway, 1 *NPR1*, 4 *TGA* and 8 *PR1* were identified (Figure 9). Twelve randomly selected genes of hormone signal transduction pathways, including 1 *AUX1*, 1 *ARF*, 2 *SnRK2*, 1 *AREB/ABF*, 2 *JAZ*, 1 *MYC2*, 1 *NPR1*, 1 *TGA* and 2 *PR1*, were verified by qRT-PCR (Appendix A). The results showed that the relative expression levels were basically consistent with RNA-seq data, which supported the accuracy and reliability of transcriptome analysis results.

## 3. Discussion

Okra has important edible and medicinal value. However, due to the complexity and lack of the okra genome, it is difficult to assess the gene function of okra [28,29]. VIGS has been widely used as an effective functional genomics tool for gene function analysis [30]. NAC TFs play important roles in plant growth, development and stress responses. In this study, the role of AeNAC83 in the salt stress responses in okra was investigated. We found that the *AeNAC83*-silenced okra seedlings produced by VIGS were slightly more sensitive to salt stress (Figure 3). Further analysis found that the gene was not completely silenced, probably because VIGS sometimes silences only part of the target gene [31]. Hence, transgenic *Arabidopsis* plants over-expressing *AeNAC83* were generated to further determine the role of AeNAC83 in growth and salt stress. Over-expression of *AeNAC83* enhanced tolerance to salt stress and suppressed vegetative growth (Figure 4). Taken together, okra NAC TF AeNAC83 plays a pivotal role in mediating plant growth and defense response to salt stress.

Transcriptomic analyses showed that compared with WT-CK, there were 4285 DEGs in WT-N, while only 1360 DEGs were found in OX3-N compared with OX3-CK (Figure 5). The result showed that overexpression of *AeNAC83* resulted in the insensitivity of numerous genes to salt stress, which may be one of the reasons for enhanced resistance to salt stress. KEGG pathway enrichment analysis showed that these DEGs mainly participated in “MAPK signaling pathway-plant”, “plant-pathogen interaction” and “phenylpropanoid biosynthesis”, “starch and sucrose metabolism” and “circadian rhythm-plant” (Figure 6). In plants, the rapid activation of MAPK cascades has long been observed involved in growth and development, as well as in response to drought, salinity, wounding, heat, and cold [32]. Moreover, for all comparison groups, the phenylpropanoid biosynthesis pathway was always found to contain significant enrichment of DEGs.

The phenylpropanoid pathway that produces lignin, flavonoids, and other secondary metabolites [26], contributes to the defense and growth of plants. Fortifying cell walls by increasing their lignin content is one of the common plant defense mechanisms [33]. The 4CL, C4H and POD are core enzymes involved in the biosynthesis of flavonoids and lignin. From our analysis, the expressions of 4CL, C4H, CAD and POD-related DEGs were increased in OX3-CK and the NaCl-treated WT and OX3 samples (Figure 7A). This finding suggested that NaCl stress could cause an increase in lignin synthesis. Under normal conditions, the *AeNAC83*-overexpression plant may accumulate more lignin, which may be one of the reasons why it has stronger salt stress resistance than the wild type. In response to a variety of abiotic stress, flavonoids (including anthocyanins) play a major antioxidant role [34,35]. Flavonoids can reduce reactive oxygen species in plant tissues [36,37,38], which are usually produced by stress such as ultraviolet radiation, drought and salt stresses. We observed up-regulation of flavonoids genes in NaCl-treated seedlings (Figure 7B), indicating that the genes involved in stress metabolism are generally up-regulated. Some transcription factors have been identified to have regulatory functions in phenylpropanoid and flavonoid biosynthesis pathways [39]. Overexpression of *PvMYB4*, a suppress phenylpropanoid metabolism TF, caused a reduction in the lignin content and decreased recalcitrance in *Panicum virgatum* [40]. NAC is the second largest class of TFs in plants and has been shown as a key regulator of abiotic stresses [41]. In *Arabidopsis* and rice, *NAC* genes’ overexpression enhanced drought and salt tolerance [20,42,43]. Additionally, overexpression of *AeNAC83* significantly induced the expression of most flavonoid biosynthesis-related DEGs under normal conditions. This result was further confirmed by the experiment of anthocyanin content determination (Appendix A). These results showed that overexpression of *AeNAC83* improved the resistance of plants to salt stress, possibly by regulating the accumulation of lignin and flavonoids.

Chloroplast is the site of photosynthesis. High Na^+^ levels can destroy the structure of chloroplast, damage the membrane system of plant and seriously degrade chlorophyll, resulting in the decline of photosynthesis. In the study, the expression levels of photosynthesis-related genes were significantly down-regulated in WT and transgenic *Arabidopsis* (OX-3) after NaCl treatment and the degree of decline of genes in WT was significantly higher than that of OX-3 (Figure 8). The results showed that *AeNAC83*-overexpression transgenic plants had stronger photosynthetic capacity under salt stress, which could ensure more organic accumulation and improve the salt tolerance of plants. The same phenomenon has been observed in transgenic *Arabidopsis* overexpressing wheat *TaNAC67*, the chlorophyll content and Fv/Fm of which were higher than those of the control [44]. In addition, under salt stress, *AeNAC83*-overexpression transgenic plants have larger roots, suggesting that AeNAC83 may regulate roots to improve salt stress tolerance. Similarly, overexpression of soybean *GmNAC20* improved the tolerance to low temperature and salt and promoted lateral root formation [45]. Under high salt stress, *Arabidopsis thaliana AtNAC2* was highly expressed in roots, and lateral roots of transgenic plants overexpressing *AtNAC2* increased [15]. Compared with wild-type plants, *AeNAC83*-overexpression plants showed growth retardation, which may be due to the redistribution of energy between stress tolerance and normal growth and development, thereby improving the survival rate of plants under salt stress. This was consistent with transgenic rice overexpressing *ONAC022* [18]. 

Phytohormones, such as auxin, abscisic acid (ABA), ethylene, gibberellic acid (GA), and jasmonic acid (JA), also play central roles in salt stress response [46]. *Malus domestica* MdNAC047 induced ethylene accumulation by increasing the expression of ethylene synthesis genes *MdACS1* and *MdACO1* and TF gene *MdERF3*, enhancing tolerance to salt stress by regulating ethylene response [47]. Soybean (*Glycine Max*) GmNAC109 promoted the formation of lateral roots of transgenic *Arabidopsis thaliana* and enhanced salt stress tolerance through positive regulation of auxin response gene *AtAIR3* and negative regulation of TF *AtARF2* [48]. To reveal the involvement of these hormones in the salt stress response and growth, numerous genes related to these hormones signaling pathways were identified (Figure 9). For example, in the ABA-signaling pathway, the core of salt- and drought-stress responses in plants [49], 1 *SnRK2* gene and 2 *AREB/ABF* genes were up-regulated and 1 *PYR* gene, 3 *PP2C* genes, 1 *SnRK2* gene and 1 *AREB/ABF* gene were down-regulated compared with WT-N. The expressions of JA- and SA-signaling pathways related to DEGs were increased in NaCl-treated WT and OX3 samples. These data indicated that AeNAC83 may enhance tolerance to salt stress by regulating various hormone signaling responses.

Interestingly, besides the increase in salt sensitivity in *AeNAC83*-silenced okra seedlings and salt tolerance in *AeNAC83*-overexpression transgenic *Arabidopsis*, the *AeNAC83*-silenced seedlings showed enhanced plant growth, while *AeNAC83*-overexpression lines exhibited the opposite phenotype. These data imply that AeNAC83 participates in the balance between defense responses and plant growth. In plants, the trade-off between defense and growth has attracted considerable attention, where enhanced resistance often impairs growth and development. For example, in rice, over-expression of *OsRCI*-*1* resulted in an increase in BPH resistance and a reduction in thousand-grain weight [50]. OsNAC2 negatively regulates root growth [51] and mediates abiotic stress tolerance [52].

## 4. Materials and Methods

### 4.1. Multiple Sequence Alignment and Phylogenetic Analysis 

The amino acid sequences of NAC proteins were obtained from the TAIR (https://www.arabidopsis.org/) and NCBI. The accession number and references for other NACs: *Arabidopsis thaliana*, ATAF1 (AT1G01720), ATAF2 (AT5G08790) [4], CUC1 (AT3G15170), CUC2 (AT5G53950) [5], CUC3 (AT1G76420) [53], AtNAC1 (AF198054.1) [14], AtNAC2 (AT5G39610) [15], AtNAP (AJ222713) [54]; rice, OsNAC2 (Os04g0460600) [51,55]; soybean, GmNAC20 (EU440353.1) [45]; petunia, PhNAM (X92204) [3]. Multiple sequence alignment of AeNAC83 with other 11 NAC members was carried out using ClustalX 2.0 and modified with GeneDoc. The phylogenetic tree was generated using the MEGA7 software with the neighbor-joining (NJ) method, Poisson-corrected distances and 1000 replicates.

### 4.2. RNA Isolation and qRT-PCR 

Mature seeds of okra (*Abelmoschus esculentus* L.) Cultivar “xian zhi” was used in this study. Plant growth conditions and treatment were performed according to our previous research [25]. Okra seedlings with only two true leaves under normal conditions were selected for salt treatment, irrigated with 300 mM NaCl solution, and the control was irrigated with water. At 1 d, 3 d, 5 d and 7 d after treatment, RNA was extracted from the second true leaf, and the expression of the *AeNAC83* gene was analyzed by qRT-PCR. The primers were listed in Appendix A.

### 4.3. Subcellular Localization of GFP-AeNAC83 Fusion Protein

The coding sequence of AeNAC83 was amplified by RT-PCR with primers 5′-GAGCTCATGGAGAAGCTTAGTTTTGT-3′ and 5′-TCTAGAAGGTTTTCTTCTGAAGTAAG-3′, and cloned into the SacI/XbaI sites of pCAMBIA1300-sGFP under the control of the CaMV 35S promoter, resulting in 35Sp:: AeNAC83:GFP construct. The constructed 35Sp:: AeNAC83:GFP vector was introduced into *Agrobacterium* strain GV3101, and transiently transformed in tobacco epidermal cells. The empty vector was used as a negative control. The distribution of fluorescence was imaged 48 h after the inoculation by a confocal laser-scanning microscope (Zeiss LSM 710).

### 4.4. Virus-Induced Gene Silencing (VIGS) of AeNAC83 and NaCl Treatment

A specific sequence of about 300bp from AeNAC83 was amplified by RT-PCR with primers 5′-GAATTCATGGAGAAGCTTAGTTTT-3′ and 5′-GGTACCTTTCCTGTCGATTCCGGT-3′, and cloned into the EcoRI/KpnI sites of pTRV2, resulting in TRV-NAC83 construct. The construct was transferred into *Agrobacterium tumefaciens* GV3101.

A single colony of the *Agrobacterium tumefaciens* containing TRV1, TRV-*NAC83* or empty TRV2 was inoculated into 3ml YEP liquid medium with 50 mg/L rifampicin and 50 mg/L kanamycin, and cultured at 28 °C for 12 h at 200 rpm. For secondary activation, 1 mL culture was added to 50 mL of YEP and grown until OD600 reached 1–1.5. Agrobacteria were resuspended in a monomethylamine (MMA) solution (20 µM acetylsyringone, 10 mM MgCl_2_ and 10 mM MES, PH 5.6) to a final concentration of OD600 = 1 and placed without shaking at 28 °C for 3 h in dark. Then the cotyledons of okra seedlings that cotyledons were fully stretched and the true leaves had not yet grown were used to inoculate with the suspension of TRV1 and TRV-*NAC83* mixed at a ratio of 1:1 (v/v). pTRV2 and pTRV1 empty vectors were used as negative controls. The inoculated seedlings were maintained for 12 h in the dark and employed to a 16-h light/8 h dark cycle at 25 °C in a growth chamber. After 25 days, the *AeNAC83* mRNA level was measured and the phenotype of leaves was observed. Then the *AeNAC83*-silenced okra seedlings and the negative control were irrigated with 300 mM NaCl solution for 7 days. The leaf fresh weight and total chlorophyll content was measured [25].

### 4.5. Generation of AeNAC83 Transgenic Arabidopsis Plants

The *Agrobacterium* GV3101 containing the 35Sp:: *AeNAC83*:GFP construct was transformed into *Arabidopsis thaliana* ecotype Columbia (wild-type, WT) via the inflorescence infiltration method. Transformed lines were selected on 1/2 MS medium with 30 mg/L hygromycin and then confirmed by PCR. Two *AeNAC83* homozygous transgenic lines (OX-3 and OX-7) of the T4 generations with high *AeNAC83* expression were selected for further phenotypic analysis. The primers were listed in Appendix A.

### 4.6. Performance of Transgenic Lines under Salt Stress Treatment

Seeds of WT and OX-3 and OX-7 were kept at 4 °C for 3 days and then plated on 1/2 MS medium under a 16-h light/8-h dark cycle at 23 °C in a growth chamber. Four-day-old seedlings were transplanted on 1/2 MS medium containing 0, 120 or 150 mM NaCl. After 10 days, the total fresh weight and primary root length were measured. Samples of whole plants of OX3 and WT treated with 0 or 120 mM NaCl were harvested and frozen in liquid nitrogen for Illumina sequencing assay. Seedlings were sown on 1/2 MS medium without NaCl treatment for two weeks, and the roots were imaged using a stereo microscope.

### 4.7. RNA Isolation and Illumina Sequencing

Total RNA was extracted from the leaves and roots of 14-day *AeNAC83*-overexpression transgenic (OX3) and the wild (WT) *Arabidopsis* treated with 120 mM NaCl. RNase-free DNase I (Takara, Dalian, China) was used to remove the DNA. cDNA libraries were constructed as described by our previous study [25]. The constructed libraries were sequenced on the Illumina platform. The resulting reads (clean reads) were mapped to the Arabidopsis reference genome using HISAT2. FPKM was used to estimate gene expression levels. Differential expression analysis of two groups was performed using the DESeq2 with an adjusted *p*-value < 0.05.

### 4.8. Gene Annotation and Enrichment Analysis

Based on a Wallenius non-central hyper-geometric distribution [56], GOseq R packages were used to analyze GO enrichment of DEGs. KOBAS software [57] was used to test whether DEGs were statistically enriched in KEGG pathways.

### 4.9. Measurement of Anthocyanin Content

Anthocyanins were extracted from seedlings of WT and OX3 grown in 1/2 MS medium supplemented with 0 or 120 mM NaCl. Leaf material from each treatment was weighed W (g), and then 1 mL of acidic methanol containing 1% HCl (*v*/*v*) was added and kept in the dark at 4 °C for 24 h. The leaching solution was centrifuged at 13,000 rpm for 10 min, and the absorbance was measured at the wavelength of 530 and 657 nm, respectively. The relative content of anthocyanins was calculated by the formula: Q_anthocyanins_ = (A_530_ − 0.25 × A_657_)·g^−1^ FW [58].

## 5. Conclusions

In this study, we demonstrate that the nucleus-located AeNAC83 participates in salt stress tolerance in okra. Additionally, AeNAC83 negatively regulates plant growth, indicating a possible node of trade-off between okra resistance and growth. Transcriptome analysis revealed that most of the genes involved in flavonoid biosynthesis and photosynthesis were up-regulated and down-regulated, respectively. Various plant hormone signaling pathway-related DEGs were also identified. Our study provides a comprehensive understanding of the molecular mechanism of AeNAC83 involved in plant growth and salt tolerance.

## Figures and Tables

**Figure 1 ijms-23-10182-f001:**
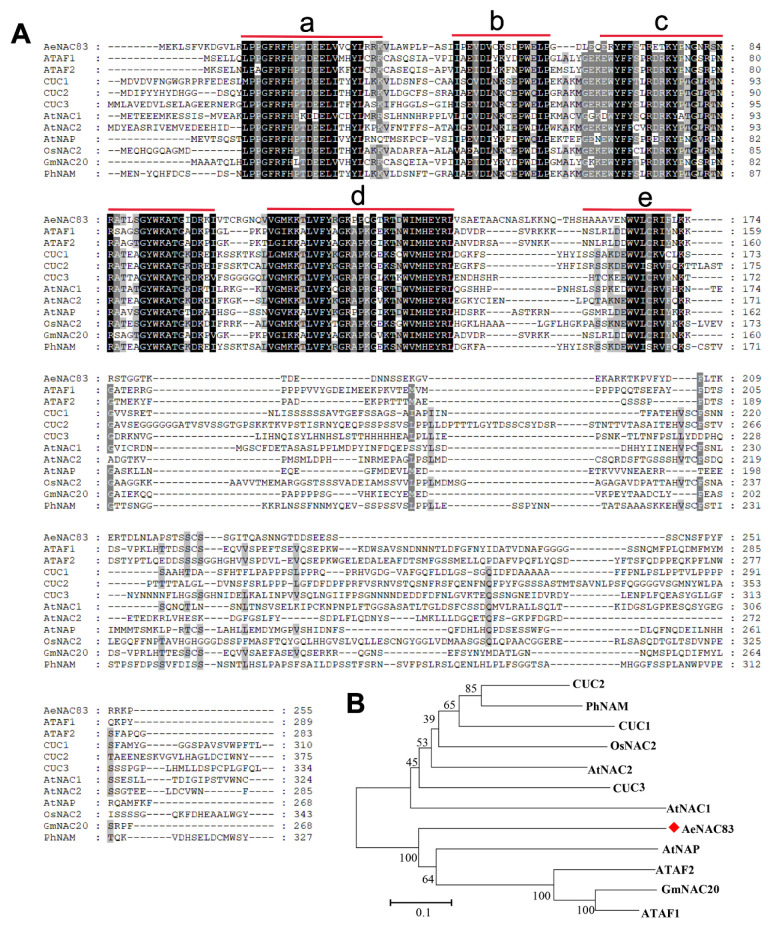
Characterization of AeNAC83: (**A**) Multiple sequence alignment of AeNAC83 and its homologous NAC proteins. The NAC domain with five conserved regions (a–e) are indicated by red lines; (**B**) Phylogenetic analysis of AeNAC83 with its homologous NAC proteins.

**Figure 2 ijms-23-10182-f002:**
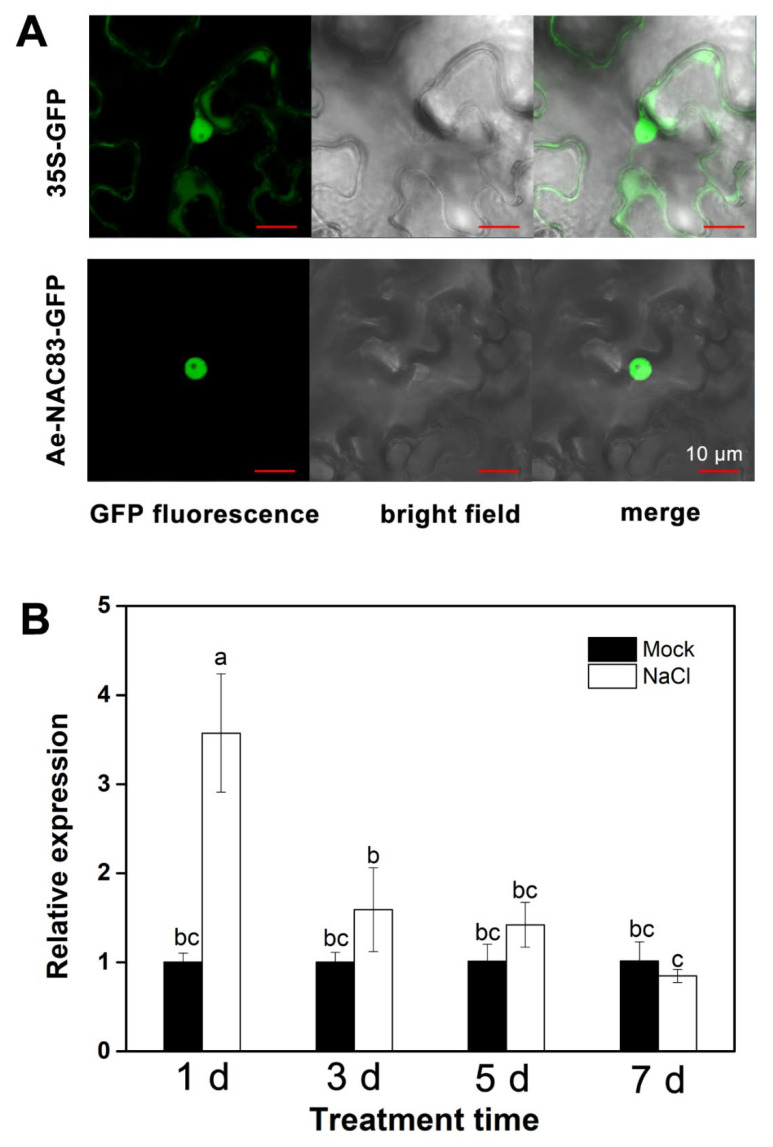
Subcellular localization and expression pattern of AeNAC83: (**A**) Subcellular localization of AeNAC83 in *N. benthamiana*. *N. benthamiana* cells were transformed with 35Sp:: *AeNAC83*:GFP or pCAMBIA1300-GFP. After incubating for 48 h, the transformed cells were observed under a confocal microscope. The photographs were taken under detecting GFP fluorescence, bright field, and in combination (merge), respectively. Empty vector (pCAMBIA1300-GFP) was used as a control; (**B**) Expression of *AeNAC83* in okra seedlings after 300 mM NaCl treatment by qRT-PCR. Total RNA for expression analysis was isolated from leaves of two-week-old seedlings after 300 mM NaCl treatment for 1, 3, 5, and 7 days. Data are presented as mean ± SD (*n* = 3). Different letters denote significant differences at *p* < 0.05, using ANOVA and Duncan’s multiple tests.

**Figure 3 ijms-23-10182-f003:**
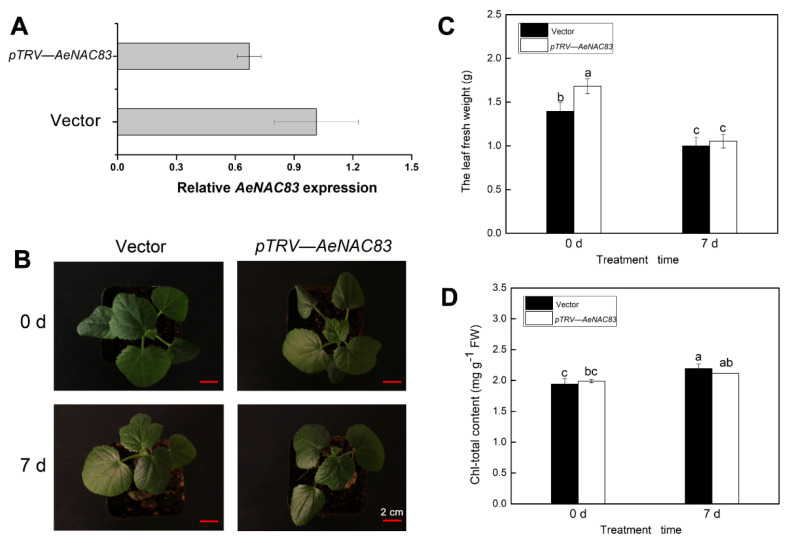
Phenotype analysis of *AeNAC83*-silenced okra plants produced by VIGS under salt stress. The *Agrobacterium tumefaciens* GV3101 cell culture harboring the pTRV2 or *AeNAC83*-pTRV2 together with pTRV1 were mixed with a ratio of 1:1 and syringe-infiltrated into okra cotyledons. At 25 d post-inoculation (dpi), the leaves were used for gene expression assay and 300 mM NaCl treatment for 7 days. (**A**) Expression analysis of *AeNAC83* in *AeNAC83*-silenced okra seedlings by qRT-PCR. (**B**) Images of plant phenotype. (**C**) The leaf fresh weight. (**D**) Total chlorophyll content. Data are presented as mean ± SD (*n* = 10). Different letters denote significant differences at *p* < 0.05, using ANOVA and Duncan’s multiple tests.

**Figure 4 ijms-23-10182-f004:**
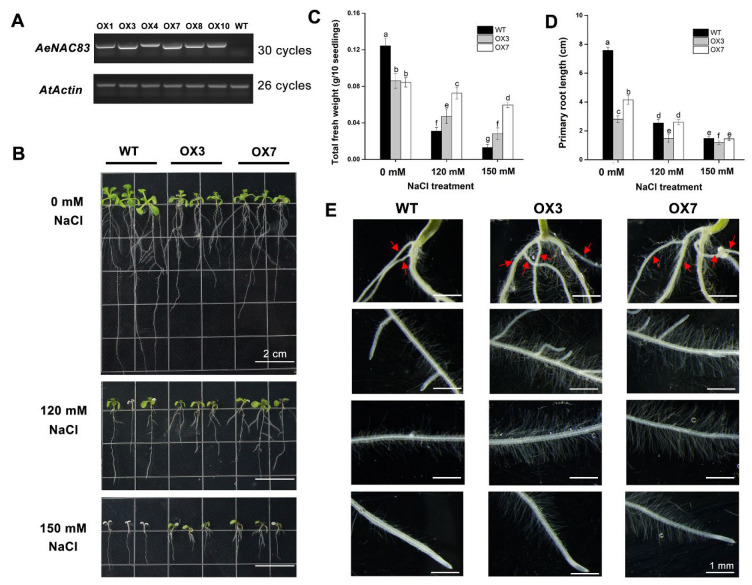
Roles of AeNAC83 in salt tolerance and in growth in *Arabidopsis* transgenic plants. Four-day-old seedlings of wild-type (WT) and two transgenic lines (OX3 and OX7) were transplanted on 1/2 MS medium supplemented with 0, 120, or 150 mM NaCl for 10 days. (**A**) Semi-quantitative analysis of *AeNAC83* gene expression in wild-type (WT) and *AeNAC83*-overexpression transgenic plants. The first bands show *AeNAC83* gene expression (30 cycles) and the bands below show *AtActin* gene expression (26 cycles) used as internal control. (**B**) Phenotypes of seedlings on 1/2 MS medium supplemented with 0, 120, or 150 mM NaCl, respectively. (**C**,**D**) Fresh weight and primary root length of seedlings at the end of the experiment in (**B**). Data are presented as mean ± SD (*n* = 10). Different letters denote significant differences at *p* < 0.05, using ANOVA and Duncan’s multiple tests. (**E**) Root phenotype of seedlings on 1/2 MS medium without NaCl treatment. The red arrows indicated the crown root.

**Figure 5 ijms-23-10182-f005:**
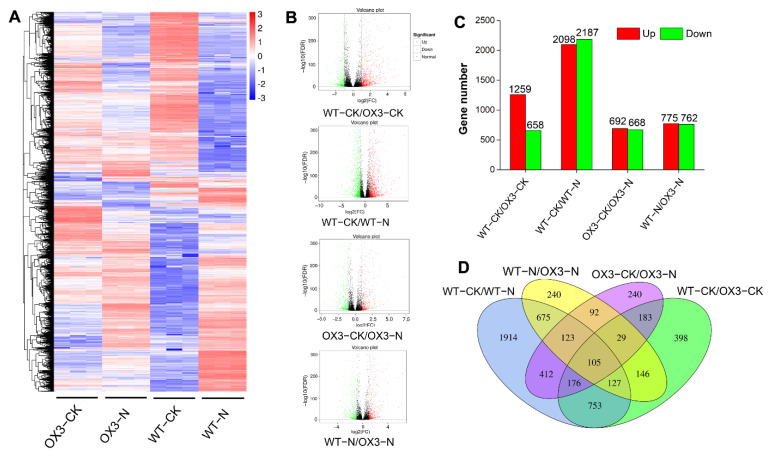
Transcriptional variations in *Arabidopsis* wild-type (WT) and transgenic plant (OX3) under NaCl treatment: (**A**) Expression profiles of the DEGs under NaCl treatment were shown by a heatmap; (**B**) Significance analysis of the DEGs in different comparisons by volcanoplots; (**C**) The number of up- and down-regulated genes in different comparisons; (**D**) Venn diagrams showed the proportions of the up- and down-regulated genes in four comparisons. WT-CK, OX3-CK: WT and OX3 grown under optimum conditions; WT-N, OX3-N: WT and OX3 were subjected to salt stress. Three replicates for each treatment.

**Figure 6 ijms-23-10182-f006:**
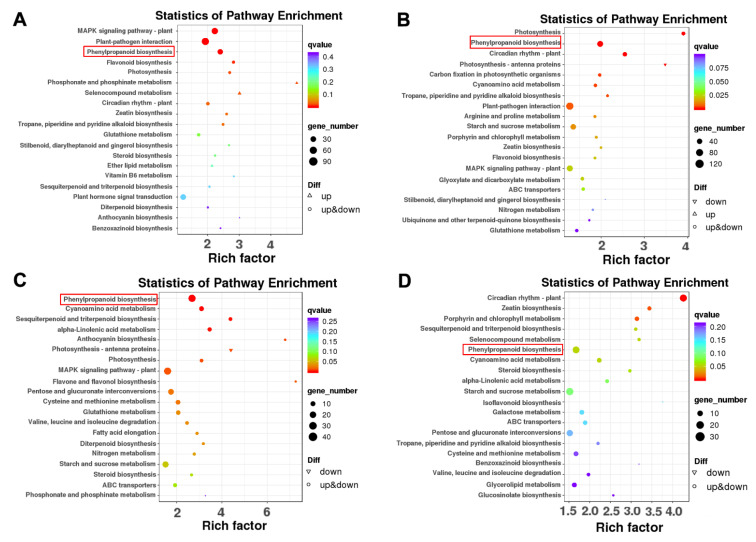
The top 20 enriched KEGG pathways enrichment analysis of DEGs in the four comparison groups. (**A**–**D**) KEGG pathway enrichment analysis of DEGs in the WT-CK vs. OX3-CK, WT-CK vs. WT-N, OX3-CK vs. OX3-N, and WT-N vs. OX3-N comparisons, respectively.

**Figure 7 ijms-23-10182-f007:**
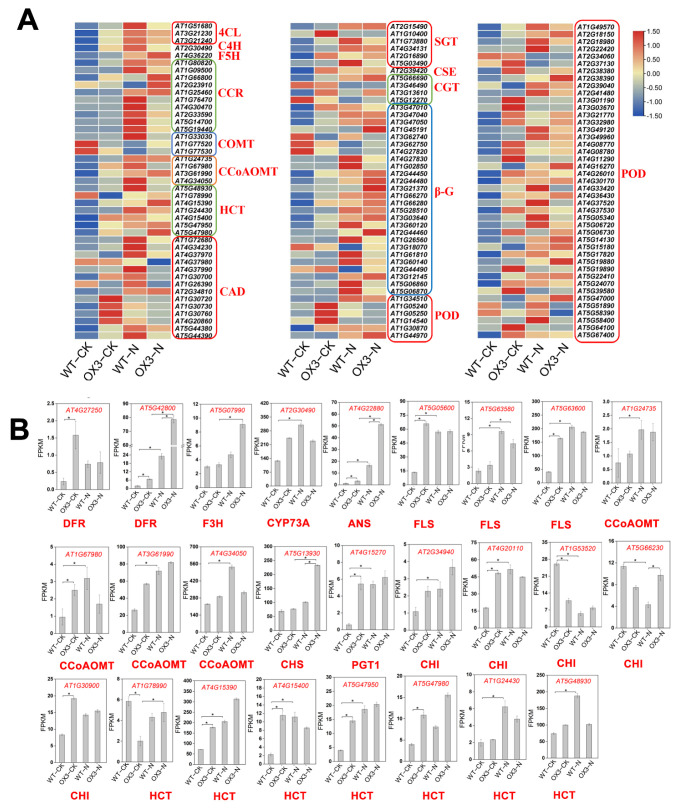
Transcript abundance changes of the phenylpropanoid and flavonoid biosynthesis pathway-related DEGs in *Arabidopsis* wild type (WT) and transgenic plant (OX3) under NaCl treatment. (**A**) Heat map of DEGs in phenylpropanoid biosynthesis pathway. The log2-transformed FPKM values of DEGs were used to generate the diagram. 4CL, 4-coumarate-CoA ligase; C4H, cinnamate-4-hydroxylase; F5H, ferulate-5-hydroxylase; CCR, cinnamoyl-CoA reductase; CCoAOMT, caffeoyl-CoA O-methyltransferase; COMT, caffeic acid 3-O-methyltransferase; HCT, hydroxyl cinnamoyl transferase; CAD, cinnamyl-alcohol dehydrogenase; SGT, scopoletin glucosyltransferase; CSE, caffeoylshikimate esterase; CGT, coniferyl-alcohol glucosyltransferase; β-G, beta-glucosidase; POD, peroxidase. (**B**) Gene expression of DEGs in flavonoid biosynthesis pathway. DFR: dihydroflavonol 4-reductase; F3H: flavanone 3-hydroxylase; CYP73A: trans-cinnamate 4-monooxygenase; ANS: anthocyanidin synthase; FLS: flavonol synthase; CHS: chalcone synthase; PGT1: phlorizin synthase; CHI: chalcone isomerase; HCT: shikimate O-hydroxycinnamoyltransferase. * indicate statistical significance based on two-tailed Student’s *t*-test at *p*-values < 0.05.

**Figure 8 ijms-23-10182-f008:**
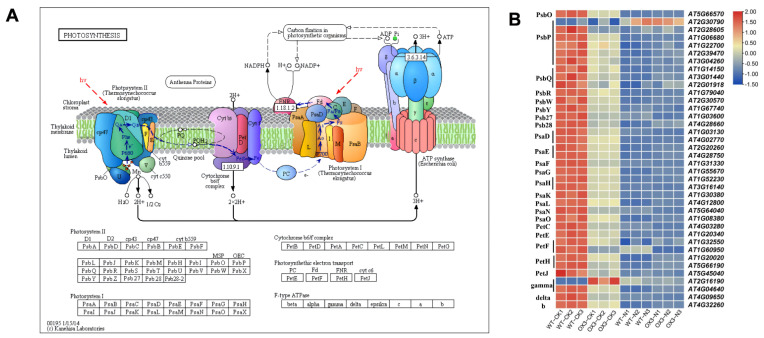
Transcript abundance changes of photosynthesis pathway-related DEGs in *Arabidopsis* wild-type (WT) and transgenic plant (OX3) under NaCl treatment. (**A**) Photosynthesis pathway (ko00195). Different letters indicated the different subunits of photosynthetic complexes. (**B**) Heat map of DEGs in photosynthesis pathway. The log2-transformed FPKM values of DEGs were used to generate the diagram.

**Figure 9 ijms-23-10182-f009:**
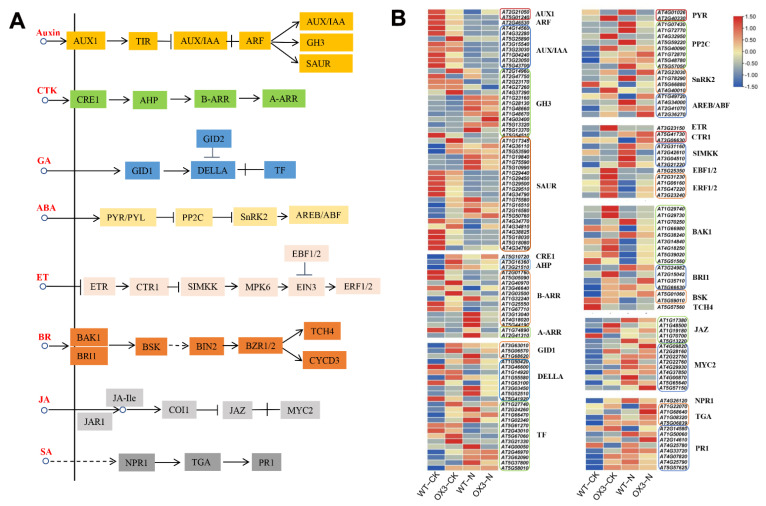
Transcript abundance changes of the various hormone signal transduction pathways-related DEGs in *Arabidopsis* wild-type (WT) and transgenic plant (OX3) under NaCl treatment. (**A**) Plant hormone signal transduction pathways. (**B**) Heat map of DEGs in various hormone signal transduction pathways. The log2-transformed FPKM values of DEGs were used to generate the diagram.

## Data Availability

The transcriptional data was deposited in the NCBI Sequence ReadArchive (BioProject: PRJNA857503). All data generated or analysed during this study are included in this published article and its Appendix A files.

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
