# Peer review of "Ectopic Expression of AeNAC83, a NAC Transcription Factor from Abelmoschus esculentus, Inhibits Growth and Confers Tolerance to Salt Stress in Arabidopsis"

_ijms, 2022, doi:10.3390/ijms231710182_

Round 1
Reviewer 1 Report
General comments
The authors have provided a good article regarding NAC genes in okra, which in my perspective this manuscript has brought out important and highly elaborative data in this manuscript. Not much detail has been described regarding NAC genes in this species. As okra is also a prospective crops in some country, therefore understanding the molecular mechanism to improve their abiotic stress tolerance is important. For this manuscript specifically, few improvements are needed from my side to fledge this article.
Specific Comments
Title: I understand that the authors want to describe okra's NAC gene here, but without any comma or parentheses for the AeNAC83 here, will make the description seems redundant. I suggest to put AeNAC83 inside parentheses.
Introduction: I suggest the authors adding a bit more details of why NAC TF gene is highlighted in okra. There are many other pathways related to salt stress mechanism, e.g., betaine aldehyde, glycine betaine, CAMTA, and more. Authors can perhaps explain briefly, perhaps using comparative data about salt stress study in other mechanism (in paragraph 2 or 3).
Results:
Figure 8. Please indicate the KEGG pathway ID in the figure legend.
Figure 9. Just my opinion, but the color pallete and gradient used here are a bit similar to each other. For instance, between CTK, ET and JA, and between GA and SA
Reviewer 2 Report
The work submitted for review was thoughtfully written.
Please check your work for spelling and punctuation. Explanations of abbreviations should appear when it is first placed in the work. This includes the abbreviation ''WT''.
Reviewer 3 Report
This manuscript has characterized the effect of AeNAC83 overexpression on salt stress response in Arabidopsis. Authors have showed that AeNAC83 overexpression inhibits plant growth but confers salt tolerance and examined transcriptomic changes in response to salt stress in WT and OE. The results may be important for agricultural application, but scientifically looks too preliminary. Here, I raised some concerns to be addressed or corrected before publication as below.
I have a serious concern on this work. This manuscript only looked at the artificial effects of ectopic overexpression of NAC83 in Arabidopsis. Since it is an okra gene, it should be extremely important to examine how NAC83 functions on okra.
Photographs in Fig. 3B look Nicotiana benthamiana not Okra. So, I suspected that all of the data in Figure 3 may be derived from N. benthamiana? If not, authors should provide Okra photos.
Transcriptome analysis has revealed that expression of the genes in phenylpropanoid-related, photosynthesis-related and phytohormone-related pathways were altered by AeNAC83-OE and salt stress. To validate alternation of them, authors should measure and compare amount of some final products in the pathways such as anthocyanin, SA and JA and discuss the obtained result.
The texts in ALL figures are too small to read. For example, numbers inside venn diagram cannot be read. They should be larger.
In Fig. 5 legend, please show how many replicates of RNA-seq were performed.
The current title may be inappropriate. It should be changed to something more in line with the content like “Ectopic expression of AeNAC83 confers tolerance to salt stress in Arabidopsis” or others.
Round 2
Reviewer 3 Report
Unfortunately, most of the points I raised have not been attempted. The amount of final products of synthetic systems with variable expression should be examined. Some letters in figures are still small. Additionally, I doubt that VIGS in okra was working well. So, authors should examine systemic infection of TRV or the virus vector in Okra.
Round 3
Reviewer 3 Report
You responded well to my request and I am satisfied.